# Assessment of Developmental Prosopagnosia in an Individual with Tourette Syndrome and Attention Deficit Hyperactivity Disorder: A Case Report

**DOI:** 10.3390/brainsci15010056

**Published:** 2025-01-10

**Authors:** Petter Espeseth Emhjellen, Randi Starrfelt, Rune Raudeberg, Bjørnar Hassel

**Affiliations:** 1Department of Neurohabilitation, Oslo University Hospital, 0424 Oslo, Norway; petemh@ous-hf.no (P.E.E.); bjornar.hassel@medisin.uio.no (B.H.); 2Department of Psychology, University of Copenhagen, 1172 Copenhagen, Denmark; 3Department of Biological and Medical Psychology, Faculty of Psychology, University of Bergen, 5007 Bergen, Norway; rune.raudeberg@uib.no

**Keywords:** prosopagnosia, neuropsychological assessment, comorbidity, heredity

## Abstract

Background/Objectives: Prosopagnosia is the inability to recognize people by their faces. Developmental prosopagnosia is the hereditary or congenital variant of the condition. The aim of this study was to demonstrate the assessment of developmental prosopagnosia in a clinical context, using a combination of commercially available clinical assessment tools and experimental tools described in the research literature. Methods: We conducted a comprehensive neuropsychological assessment of a man with Tourette syndrome and attention deficit hyperactivity disorder (ADHD). The patient (ON) had experienced difficulties with face identity recognition throughout his life but believed they were caused by a lack of interest in others. Results: The neuropsychological assessment revealed varying degrees of difficulties primarily related to executive functions, attention, reaction time, and memory processes, as expected in a person with Tourette’s syndrome and ADHD. In addition, ON reported severe problems with face recognition on a prosopagnosia questionnaire and demonstrated severely impaired performance on tests of face memory and face perception commonly used to diagnose prosopagnosia. Interestingly, he reported familial face recognition problems on the maternal side of the family, while tics and ADHD symptoms occurred on the paternal side. This suggests that, in this case, the conditions were likely inherited through different genetic pathways. Conclusions: Proper assessment of face recognition problems, which includes a broad spectrum of clinical assessment tools, could help patients develop awareness and acceptance of themselves and their difficulties, and could serve as a basis for the development of clinical interventions. While ON’s DP, Tourette syndrome, and ADHD may have distinct genetic origins, impairment in face identity recognition has been observed across several neurodevelopmental conditions and is likely more common than currently thought.

## 1. Introduction

Prosopagnosia, or face blindness, is characterized by severely impaired face identity recognition due to disturbances in face processing mechanisms [1]. Prosopagnosia following brain injury (acquired prosopagnosia) is well documented following focal lesions in occipital and temporal areas in the right hemisphere or bilaterally [2]. More recently, it has become clear that prosopagnosia also exists in a congenital or developmental form (developmental prosopagnosia; DP) [3].

Face perception and recognition are supported by a distributed neural network where portions of the inferior occipital gyrus, the fusiform gyrus, and the medial temporal lobe play crucial roles [4]. The right hemisphere is dominant for face processing in most people, but regions in the left hemisphere also contribute to efficient face processing, and acquired prosopagnosia is typically more severe following bilateral lesions [2]. DP has been estimated to affect around 2–3% of the population [5,6,7], although both diagnostic criteria, cut-off scores, and prevalence estimates are continuously debated within the field (see for example [8,9,10,11]).

Face recognition may be impaired in different neurodevelopmental and psychiatric conditions, e.g., autism spectrum disorder [12,13,14,15], schizophrenia [16,17], dyslexia [18,19,20], attention deficit hyperactivity disorder (ADHD) [21,22], Turner syndrome [23], and Möbius sequence [24]. Many people with face recognition problems report other co-occurring deficits, for example, perceptual or memory problems, and diagnoses [25]. Despite the increasing focus on both specific and associated impairments in face recognition, few clinical tools are available for the assessment of this important cognitive function. The aim of this report is to illustrate how face identity recognition can be assessed, informed by DP assessment guidelines [3,26], in patients with neurodevelopmental disorders with the use of both experimental and commercially available clinical assessment tools.

## 2. Materials and Methods

### 2.1. Case Description

The patient, referred to as “ON” in this report, was suspected by his general practitioner to have Tourette syndrome (TS). He was referred to the Department of Neurohabilitation, Oslo University Hospital, Oslo, Norway, where he was diagnosed with TS and ADHD, and he underwent neuropsychological assessment of face recognition difficulties. ON was then in his late forties.

In addition to motor and phonic tics, ON had had lifelong problems related to attention, impulsivity, and hyperactivity. He also reported having difficulties recognizing people by their faces. His father had a diagnosis of TS as well as ADHD-like symptoms. Further, his paternal uncle and grandfather had chronic motor and phonic tics. ON had heard anecdotes of several relatives on his mother’s side, including his maternal uncle, having faced identity recognition problems. It was his partner who, when ON was in his early forties, made him aware that he might have “face blindness”. ON did not know that face blindness, or prosopagnosia, was a recognized disorder. However, he remembered thinking about his face identity recognition difficulties from as early as the age of three and he had a lifetime experience of trying to compensate for his difficulties. For instance, he paid attention to cues such as body figures, gait, posture, or voice to identify individuals. His partner also helped him remember people that he could not recognize. He described how he used his appearance (having distinctive clothes and facial hair) and social roles (he had a prominent position in a local voluntary organization) to attract other people’s attention. Thus, by making other people notice and approach him, instead of him having to address them, cues to their identity other than facial features could be revealed. ON reported experiencing embarrassment and shame because of his face identity recognition problems. He recounted several embarrassing episodes involving problems recognizing people. On one occasion, he mistook a mirror image of his face for a stranger staring at him. On another occasion, he failed to recognize his own son when they unexpectedly met at a shopping center. For a long time, he had assumed that his poor face recognition ability was caused by a lack of interest in others.

At the time of assessment, ON had completed 12 years of formal education, equivalent to completing high school in the U.S. He had had several jobs and currently held a managerial position in a small non-governmental organization. ON reported difficulties with short-term memory and trouble remembering instructions or information that did not interest him. However, his inability to recognize people by their faces stood out to him as a specific memory deficit. He did not report reading or spelling difficulties. Nor did he report topographical orientation difficulties, which have a high prevalence in people with prosopagnosia [27,28]. Apart from the tics, a standard neurological examination identified the left hand as dominant. He had normal sensory function, eyesight, visual field, eye movement, and color vision (assessed by Ishihara’s tests for color deficiency [29]). A 1.5 T MRI examination (T1- and T2-weighted images, including fluid-attenuated inversion recovery and diffusion-weighted images) revealed no structural abnormalities of the brain, including the forebrain, cerebellum, and brainstem. A report from an optician indicated only mild hyperopia. An in-depth psychiatric assessment did not reveal any additional psychiatric conditions.

### 2.2. Neuropsychological Assessment

A thorough neuropsychological assessment, informed by DP assessment guidelines [3,26] and previous studies of DP (e.g., [28,30,31]), was conducted by a neuropsychologist in training (the first author) under the supervision of an experienced board-certified neuropsychologist. Scores on commercially available clinical assessment tools were compared to norms from official manuals. For non-commercial assessment tools normative samples from the literature, described in more detail below, were used. Official Norwegian translations of assessment measures were used where available.

General cognitive ability, including verbal comprehension, perceptual reasoning, working memory, and processing speed, were assessed using the Wechsler Adult Intelligence Scale—Fourth Edition (WAIS-IV) [32]. Low-level visual perception was assessed using the Stars Cancellation Test [33], six subtests from the Birmingham Object Recognition Battery (BORB) [34] and all subtests of the Visual Object and Space Perception Battery (VOSP) [35]. Visuospatial memory was assessed using the Brief Visuospatial Memory Test—Revised (BVMT-R) [36], the Continuous Visual Memory Test (CVMT) [37], and the Spatial Span subtest from the Wechsler Memory Scale—Third Edition (WMS-III) [38]. Verbal memory was assessed using the California Verbal Learning Test—Second Edition (CVLT-II) [39] and the Logical Memory subtests from the WMS-III [38]. Executive and attentional control functions were assessed using the Conners Continuous Performance Test—Third Edition (CCPT-3) [40], the Tower Test from the Delis–Kaplan Executive Function System (D-KEFS) [41], and the Behavior Rating Inventory of Executive Function—Adult Version (BRIEF-A) [42]. Social cognitive functioning was assessed using the Social Responsiveness Scale—Second Edition (SRS-2) [43].

Face recognition memory was assessed using the Cambridge Face Memory Test (CFMT) [44] and the Australian CFMT (CFMT-Aus) [45]. CFMT requires learning and recognition of six cropped faces (excluding hairlines) among distractor faces and under different conditions, such as varying viewpoints, levels of lightening, and levels of visual noise [44]. The CFMT-Aus is equivalent to the CFMT in design, but the latter uses Australian faces that are more similar to Northern European faces [45]. For the CFMT, we used data available from Gerlach et al. [46]. These data are from a Scandinavian (Danish) control sample of 61 individuals (40 female), and match ON fairly well in terms of age (mean age = 36.7 years; *SD* = 9.8; range 16–56) and education (mean education = 16 years; *SD* = 1.3; range 10–17). Norms for the CFMT-Aus were taken from an Australian sample of 34 young males (whole sex-mixed sample aged 18–32 years) [45].

Because prosopagnosia may affect face memory as well as perception, we used the Cambridge Face Perception Test (CFPT) [47] to assess face perception abilities. It requires subjects to rank a series of six faces for the degree of similarity to a target face in each trial. The six face images have been morphed with other faces to obtain varying degrees of similarity to the target face. The CFPT is thus a test of face perception without demands on memory. Scores are computed by summarizing each face’s deviations from the correct position (e.g., two positions away = two errors) [47]. For the CFPT we applied reference data from Gerlach et al. [46].

We used the self-report questionnaire 20-Item Prosopagnosia Index (PI-20) to assess DP severity [48]. For the PI-20, we used norms from 319 healthy adults aged 18–74 years [48].

The two CFMTs, the CFPT, and the PI-20 are considered leading diagnostic tests with evidence of good psychometric properties [3,7,26,27,49,50].

Within-class [47] object recognition memory was assessed using the Cambridge Car Memory Test (CCMT) [51]. This test is matched in format to the CFMT but uses cars as stimuli instead of faces. The CCMT and the CFMT thus provide a direct comparison between recognition memory for faces versus cars (cars are considered a proxy for object recognition) [52]. For the CCMT we used reference data from Gerlach et al. [46].

Qualitative descriptive labeling of performance test scores follows the American Academy of Clinical Neuropsychology consensus conference statement, which prescribes different labeling depending on whether normative scores are near-normally or non-normally distributed [53]. For questionnaires, scores are labeled exceptionally high (>2 *SD* above the normative mean), high (>1 *SD* above the normative mean), or within normal levels (within 1 *SD* above the normative mean).

### 2.3. Single Case Analysis

In order to further explore the severity and specificity of ON’s face recognition deficits in relation to the recognition of other visual objects, his scores were compared to the controls’ mean scores on the CFMT, CCMT, and CFPT using single case statistics developed by Crawford and colleagues [54,55]. We tested whether ON’s performance in each individual test was significantly impaired compared to controls using the program Singlims_ES (available here https://homepages.abdn.ac.uk/j.crawford/pages/dept/Single_Case_Effect_Sizes.htm, accessed on 16 November 2024). To determine if his face recognition was selectively impaired, i.e., if there was a dissociation between performance with faces and other objects, we compared performance on the CFMT and CCMT to controls, using the BTSD test for dissociation as implemented in the program DissocBayes_ES (available at the website linked to above). This analysis tests whether the difference in ON’s performance on the two tasks exceeds the difference in performance observed in the control group while taking the correlation between tasks in the control group into account. SPSS Version 29.0.1.0 was used to calculate means and *SD*s used in the single case analyses.

## 3. Results

### 3.1. Neuropsychological Profile

ON’s scores on neuropsychological performance-based tests and self-report questionnaires are shown in Table 1.

Most scores were within normal expectations (for tests with non-normal distributions) or average or higher (for tests with near-normal distributions). ON obtained low average and below average scores on specific tests of verbal working memory under high attentional demands (first trials of list A and B from the CVLT-II), visual perception (VOSP Object decision), visual recognition memory (CVMT Total Score, CCMT), and reaction time and attention (CCPT-3 Hit Reaction Time, Hit Reaction Time *SD*, and Hit Reaction Time Interstimulus Interval Change). ON reported exceptionally high levels of executive dysfunction on the BRIEF-A. He reported normal social cognitive responses on the SRS-2, which was consistent with the clinical impression that he did not have an autism spectrum disorder. He obtained exceptionally low scores on a test of face perception (CFPT) and on two tests of face memory (CFMT; CFMT-Aus). His responses on the PI-20 qualified for severe DP [48].

### 3.2. Single-Case Analysis

Comparison of ON’s scores on CFMT, CCMT, and CFPT compared to controls from Gerlach et al. [46] are presented in Table 2.

In sum, the analyses revealed that ON’s performance on the CFMT was significantly impaired (*z* score = −3.49) compared to controls, while his performance on the CCMT was within the normal range (*z* score = −0.99). The difference between the two scores was significant (RSDT *t* value = 1.807, *df* = 60, *p* [one-tailed] = 0.038; the correlation between CFMT and CCMT in the control group = 0.074). This indicates a classical dissociation between ON’s performance on the two tests, meaning that his performance on the CFMT was selectively impaired compared to his within-range performance with cars. ONs performance on the CFPT was significantly impaired compared to the control group, indicating that ON’s trouble with faces was not merely a memory problem but also included face perception.

### 3.3. DP Diagnosis

ON fulfilled the proposed diagnostic criteria for DP [3,26]. He had exceptionally low scores on three tests of face memory and/or perception, and he reported lifelong difficulties with remembering people’s faces. He did not show signs of visual agnosia. He showed only milder and less consistent difficulties with processing non-facial stimuli, relative to his exceptionally low scores on face processing tests. Furthermore, the assessment excluded any conditions known to be associated with face recognition impairments (e.g., autism spectrum disorder or broader, general cognitive impairment). There was no evidence of brain lesions that could explain his face identity recognition difficulties.

## 4. Discussion

We have demonstrated how face recognition difficulties can be assessed in a clinical context by a combination of commercial and non-commercial assessment tools. The findings are relevant to several ongoing debates in the field.

### 4.1. Face Recognition and Other Cognitive Functions

There is a longstanding discussion of whether recognition difficulties related to prosopagnosia are specific to faces, or whether they extend to other objects as well (see [52] and accompanying commentaries for an extensive discussion). Of note, while showing good visual perceptual and memory functioning on many measures, ON obtained low average (CVMT, CCMT) and below average (VOSP Object Decision) scores on three tests of visual processing. These low average and below average scores could be expressions of co-occurring and relatively milder visual perceptual or visual memory difficulties. Indeed, varying degrees of general visual processing deficits have been found in other people with DP [27,56]. Furthermore, if difficulties with face recognition are disproportionally larger than object recognition problems or other perceptual problems, a diagnosis of DP still holds [10]. Moreover, there is some evidence of DP sub-groups, where some show a clear dissociation between face and object recognition and others also have object recognition deficits [57]. The single-case analysis with Danish norms revealed a classical dissociation between recognition memory for faces (CFMT score) versus objects (cars, measured by the CCMT), which provides strong evidence that ON’s memory deficit was specifically related to face recognition.

In addition to face processing deficits and possibly mild visual perceptual or visual memory problems, ON showed varying degrees of difficulties in other cognitive domains, including executive functions, attention, reaction time, and verbal working memory under high attentional demands.

Some of ON’s deficits are most likely related to his TS and/or ADHD as impairment in executive functions, attention, working memory, and memory acquisition, which are common in people with TS and ADHD and might relate to frontal and frontostriatal dysfunctions [58,59,60].

### 4.2. The Importance of Proper Assessment

Prosopagnosia can lead to anxiety, embarrassment, guilt, a restricted social circle, and limited employment opportunities [61]. Indeed, ON confirmed these experiences. ON reported that learning that DP is a recognized condition made it easier for him to communicate his challenges in social interaction to others. He was also relieved to learn that his cognitive processing deficits were relatively specific to faces and expressed that he could now be more accepting of his shortcomings in facial recognition abilities. Furthermore, he reported not having realized that he had face recognition difficulties before late into adulthood. This has also been reported by other people with DP [62]. Even though many report a positive effect of disclosing their DP, most adults are reluctant to do so, fearing misunderstanding and stigma [63].

One goal of clinical neuropsychology should be testing predictions of different hypotheses (e.g., DP versus other perceptual or memory problems) that may be clinically useful to patients. For years, ON believed that his inability to remember faces was a problem of low interest in other people, that is, a problem of motivation rather than ability. Proper identification, including valid measures of face recognition abilities, could be a first step towards helping people with DP develop awareness and acceptance of their difficulties. Furthermore, proper assessment of face recognition abilities along with general neuropsychological functioning is a prerequisite for designing clinical interventions.

## 5. Conclusions

We have documented severe impairment in face identity recognition in an individual with TS and ADHD, based on thorough neuropsychological assessment using both clinical and experimental tests. While face recognition problems have been described in people with various other neurodevelopmental impairments, this is to our knowledge, the first description of a co-occurring DP, TS, and ADHD.

Case studies do of course have limited generalizability as to the specific findings, but the clinical approach used here can be generalized and applied to similar cases. When face processing deficits are suspected, we encourage clinicians to employ a sufficiently broad spectrum of tools to assess face recognition difficulties as well as other functions relevant to understanding patients’ difficulties in everyday life. While case studies are well suited to describe specific symptomatology in depth, broader population studies are needed to increase our understanding of the co-occurrence of face recognition impairment and other neurodevelopmental and psychiatric conditions more generally. Such studies may also contribute to the further development of assessment procedures.

Interestingly, ON’s account of several individuals with tics and ADHD traits on his father’s side and prosopagnosic traits on his mother’s side, point to distinct genetic origins in his case. Systematic questions about family history might thus provide clues to whether neurodevelopmental co-morbidity is caused by one or more genetic vulnerability factors.

## Figures and Tables

**Table 1 brainsci-15-00056-t001:** ON’s neuropsychological profile.

Measure	Raw Score	Norm-Based Score	Qualitative Descriptive Label
General cognitive ability			
WAIS-IV			
Fullscale IQ		111 ^a^	High average
Verbal Comprehension Index		102 ^a^	Average
Perceptual Reasoning Index		120 ^a^	Above average
Processing Speed Index		112 ^a^	High average
Working Memory Index		105 ^a^	Average
Visual perception			
BORB			
Length match	29	1.3 ^b^	Within normal expectations
Size match	27	−0.1 ^b^	Within normal expectations
Orientation match	26	0.5 ^b^	Within normal expectations
Position of gap match	34	−0.3 ^b^	Within normal expectations
Picture Naming (animate)	15	1 ^b^	Within normal expectations
Picture Naming (inanimate)	20	0.6 ^b^	Within normal expectations
VOSP			
Object perception			
Screening test	20	0.2 ^b^	Within normal expectations
Incomplete letters	20	0.9 ^b^	Within normal expectations
Silhouettes	26	0.7 ^b^	Within normal expectations
Object decision	16	−1.6 ^b^	Below average
Progressive silhouettes	9	0.3 ^b^	Within normal expectations
Space perception			
Dot counting	10	0.3 ^b^	Within normal expectations
Position discrimination	20	0.4 ^b^	Within normal expectations
Number location	10	0.6 ^b^	Within normal expectations
Cube analysis	9	−0.3 ^b^	Within normal expectations
Star Cancellation Test	54/ 54	-	-
Memory			
BVMT-R			
Trial 1	8	60 ^c^	High average
Total Recall	30	61 ^c^	High average
Delayed Recall	11	59 ^c^	High average
Recognition Discrimination	6	>16%ile	Within normal expectations
Index			
CVMT			
Total Score	75	−0.8 ^b^	Low average
Delayed Recognition	4	−0.5 ^b^	Average
Visual discrimination	7/7	−	-
CVLT-II			
Trial 1	5	−1 ^b^	Low average
List B trial 1	4	−1 ^b^	Low average
Trials 1–5 Total Score	46	52 ^c^	Average
Short Delay Free Recall	10	0.5 ^b^	Average
Long Delay Free Recall	10	0 ^b^	Average
Recognition Hits	14	0 ^b^	Average
Recognition False Positives	4	−0.5 ^b^	Average
Forced Recognition	16/ 16	-	-
WMS-III			
Logical Memory I—Recall	48	12 ^d^	High average
Logical Memory II			
Recall	30	12 ^d^	High average
Recognition	27/ 30	-	-
Spatial Span	19	14 ^d^	Above average
CCMT	43	−1 ^b^	Low average
Executive and attentional control			
BRIEF-A Self Report Form			
Behavioral Regulation Index		76 ^c^	Exceptionally high
Metacognitive Index		81 ^c^	Exceptionally high
Global Executive Composite		81 ^c^	Exceptionally high
CCPT-3			
d′	-	57 ^c^	High average
Omissions	-	55 ^c^	Average
Commissions	-	53 ^c^	Average
Perseverations	-	55 ^c^	Average
Hit Reaction Time	-	39 ^c^	Low average
Hit Reaction Time SD	-	39 ^c^	Low average
Variability	-	44 ^c^	Average
Hit Reaction Time Block	-	43 ^c^	Average
Change			
Hit Reaction Time	-	32 ^c^	Below average
Interstimulus Interval Change			
D-KEFS Tower Test Total	18	11 ^d^	Average
Achievement			
Social cognition			
SRS-2 Self Report Form	54	50 ^c^	Within normal levels
Face processing			
CFPT	66	−2.1 ^b^	Exceptionally low
CFMT	34	−3.5 ^b^	Exceptionally low
CFMT-Aus	25	−4 ^b^	Exceptionally low
PI-20	91	4.8 ^b^	Exceptionally high

Note. The following norm-based scores were applied: ^a^ IQ or index scores (*M* = 100, *SD* = 15), ^b^ *z* scores (*M* = 0, *SD* = 1), ^c^ T scores (*M* = 50, *SD* = 10), ^d^ scaled scores (*M* = 10, *SD* = 3), and percentile ranks. Norm-based scores for performance-based tests were adjusted so that lower scores indicate lower performance and higher scores indicate higher performance. For questionnaires, higher scores indicate higher symptom levels. BORB = Birmingham Object Recognition Battery; BRIEF-A = Behavior Rating Inventory of Executive Function—adult version; BVMT-R = Brief Visuospatial Memory Test—Revised; CCMT = Cambridge Car Memory Test; CFMT = Cambridge Face Memory Test; CFMT-Aus = Cambridge Face Memory Test—Australian; CFPT = Cambridge Face Perception Test; CCPT-3 = Conners Continuous Performance Test—third edition; CVLT-II = California Verbal Learning Test—second edition; CVMT = Continuous Visual Memory Test; D-KEFS = Delis–Kaplan Executive Function System; PI-20 = 20-Item Prosopagnosia Index; SRS = Social Responsiveness Scale—second edition; WAIS-IV = Wechsler Adult Intelligence Scale—fourth edition; WMS-III = Wechsler Memory Scale—third edition.

**Table 2 brainsci-15-00056-t002:** Scores for controls (*N* = 61) from [46] and ON.

	Controls Min	Controls Max	Controls Mean	Controls *SD*	ON	ON *z* Score	ON *z* Score CI	Two-Tailed *p*
CFMT	45	72	59.02	7.17	34	−3.49	(−4.16–-2.82)	0.001
CCMT	31	69	52.05	9.16	43	−0.99	(−1.29–-0.68)	0.331
CFPT	12	92	39.08	13.10	66	2.06	(1.61–2.5)	0.046

## Data Availability

The original contributions presented in this study are included in the article/Appendix A. Data availability is restricted due to patient confidentiality.

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
