# Peer review of "Assessment of Developmental Prosopagnosia in an Individual with Tourette Syndrome and Attention Deficit Hyperactivity Disorder: A Case Report"

_brainsci, 2025, doi:10.3390/brainsci15010056_

Round 1

Reviewer 1 Report

Comments and Suggestions for Authors

The article focuses on the investigation of developmental prosopagnosia (the inability to recognize faces) in a patient with Tourette syndrome and attention deficit hyperactivity disorder (ADHD). The study combines commercially available and experimental tools to assess cognitive functions and identify specific deficits in the patient.

The authors define the aim of the article as demonstrating methods for diagnosing prosopagnosia in clinical practice, particularly in the context of co-occurring neurological conditions. The study emphasizes the importance of assessing face recognition abilities and their impact on patients' psychosocial functioning.

This research seeks to address gaps in scientific knowledge regarding the insufficient evaluation of prosopagnosia in individuals with associated neurological disorders, such as Tourette syndrome and ADHD. It also explores the genetic underpinnings of these conditions and their connections to various cognitive functions.

The novelty of the article lies in its description of a clinical case featuring the co-occurrence of prosopagnosia, Tourette syndrome, and ADHD, assessed through comprehensive neuropsychological testing. The study is the first to highlight the genetic characteristics and the impact of these combined disorders on the patient's social interactions.

Minor comments:

  1. The article's objectives could be more explicitly connected to practical applications, such as developing guidelines for patients or clinicians.
  2. Tables (e.g., Table 1) could benefit from clearer explanations of certain metrics to make the data more accessible to a wider audience.
  3. While the discussion is extensive, it could include more practical recommendations for clinicians on using the presented tools in everyday practice.

Reviewer 2 Report

Comments and Suggestions for Authors

This paper by Emhjellen and colleagues aimed at investigating the assessment of developmental prosopagnosia in an individual with co-occurring Tourette Syndrome and ADHD, providing a detailed case report of a man with lifelong difficulties in recognizing faces, alongside with symptoms of TS and ADHD.

The manuscript is overall well-structured and scientific sound, and it shows its novelty addressing a gap in the literature of the field by highlighting potential genetic origins and familial patterns of neurodevelopmental disorders.

The introduction is comprehensive but with too many details that risk to  overwhelm the reader. Thus, a more concise overview of DP's neural basis and prevalence might improve accessibility.

The case is reported in a well-strucuted way, and the specific section is clear and easily understandable.

The discussion effectively merges the findings with the ongoing debate in the field, and it provides thoughtful insights into clinical and social implications.

Among the limitations, the nature of the study must be highlighted as a strong limitation about statistic generalizability. A deeper discussion of how this case aligns or contrasts with broader population studies could help improve the overall quality of the manuscript.

The bibliography is up-to-date and large enough for the paper.

Comments on the Quality of English Language

English language is overall fine, but grammar and sentence structure should be revised, as there are some typos and mistakes that make reading less clear.
